# Comparison of the glacial isostatic adjustment behaviour in glacially induced fault models

Rebekka Steffen<sup>1</sup>, Holger Steffen<sup>2</sup>, and Patrick Wu<sup>3</sup>

<sup>1</sup>Department of Earth Sciences, Uppsala University, Villavägen 16, 75692 Uppsala, Sweden
 <sup>2</sup>Lantmäteriet, Lantmäterigatan 2, 80182 Gävle, Sweden
 <sup>3</sup>Department of Earth Sciences, The University of Hong Kong, Pokfulam Road, Hong Kong

Correspondence to: Rebekka Steffen (rebekka.steffen.geo@gmail.com)

**Abstract.** We compare the glacial isostatic adjustment (GIA) behaviour of two approaches developed to model the movement of a glacially induced fault (GIF) as a consequence of stress changes in the Earth's crust caused by the GIA process. GIFs were most likely, but not exclusively reactivated at the end of the last glaciation. Their modelling is complicated as the GIA process involves different

- spatial and temporal scales and they have to be combined to describe the fault reactivation process accurately. Model approaches have been introduced by Hetzel & Hampel (2005, termed HA in this paper) and Steffen et al. (2014a, termed WU in this paper). These two approaches differ in their geometry, their boundary conditions and the implementation of stress changes. While the WU model is based on GIA models and thus includes the whole mantle down to the core-mantle boundary at
- a depth of 2891 km, the HA models include only the lithosphere (mostly 100 km) and simulate the mantle using dashpots. They further apply elastic foundations and a lithostatic pressure at the base of the lithosphere, while the WU models apply elastic foundations at all horizontal boundaries in the model with density contrasts. Using a synthetic ice model as well as the Fennoscandian Ice Sheet, we find large discrepancies in modelled displacement, velocity and stress between these approaches.
- The HA model has difficulties in explaining relative sea level curves in Fennoscandia such as the one of Ångermanland (Sweden), where differences of up to 123 m to the data (with data error of 18.7 m) result. The WU model differs by up to 11 m, but falls within the error bar of 11.6 m. In addition, the HA model cannot predict the typical velocity field pattern in Fennoscandia. As we also find prominent differences in stress, we conclude that the simulation of the mantle using dashpots
- is not recommended for modelling the GIA process. The earth model should consist of both lithosphere and mantle, in order to correctly model the displacement and stress changes during GIA. We emphasize that a thorough modelling of the GIA process is a prerequisite before conclusions on understanding GIF evolution can be drawn.

#### 1 Introduction

- Geodynamic models are developed to advance our understanding of the many individual as well as overlapping processes of the Earth. A common phenomenon is that several models co-exist for the same process and they should be compared or benchmarked in order to verify that each method works correctly. Benchmark studies thus have been performed for dedicated convection models (e.g. Zhong et al., 2008; Tosi et al., 2015), dynamo models (e.g. Christensen et al., 2001; Jackson et al.,
- 2014) or models of glacial isostatic adjustment (GIA; Spada et al., 2011). The latter describe the response of the Earth in terms of deformation as well as stress, rotation and geopotential changes due to changing ice-ocean load distributions on the Earth's surface. Among other things, the GIA model benchmark showed that the displacement results from models based on the viscoelastic normal mode method are comparable to results from spectral-finite element and finite-element (FE) models, when
- an earth model is subjected to an ice load. This is of importance as FE models are able to handle faults and lateral heterogeneities in the Earth's subsurface as well as nonlinear or composite rheologies in the mantle.

In this paper, our focus is on the GIA description in glacially induced fault (GIF) models. GIFs represent reactivated faults in or nearby formerly glaciated areas such as North America or north-

- ern Europe (e. g. Kujansuu, 1964; Lagerbäck, 1978; Quinlan, 1984; Johnston, 1987; Olesen, 1988; Dyke et al., 1991; Shilts et al., 1992; Fenton, 1994; Arvidsson, 1996; Muir-Wood, 2000; Stewart et al., 2000; Munier & Fenton, 2004; Sauber & Molnia, 2004; Lagerbäck & Sundh, 2008; Brandes et al., 2012). Even historical earthquakes of the last 1200 years in northern Germany are related to the last glaciation of northern Europe (Brandes et al., 2015). Movement of faults under the ice sheets in Lau-
- rentia and Fennoscandia was suppressed during glacial loading (Johnston, 1987), but was reactivated near the end of deglaciation (Wu & Hasegawa, 1996).

GIF modelling has been a challenging task as it involves the large spatial scale (> 1000 km) and long time scale tectonics stress (millions of years), the GIA induced stress (thousands of years) and the short-term earthquake motion (a few seconds to minutes) at a fault (of some km length). Nonethe-

- less, two approaches for GIF modelling were introduced in recent years, and both used FE techniques: the first was presented by Hetzel & Hampel (2005, hereafter denoted as HA) based on rather geological aspects and the second by Steffen et al. (2014a) based on the GIA modelling approach by Wu (2004, hereafter WU), which was part of the benchmark study of Spada et al. (2011). Hence, WU has rigorous support from other GIA modelling techniques, while HA has not, although it
- was applied in numerous GIF, but mainly parameter studies (Hampel & Hetzel, 2006; Hampel et al., 2007, 2009; Turpeinen et al., 2009; Hampel et al., 2010a, b). Therefore, our aim in this study is to compare these two approaches in terms of their description of the GIA process to verify (1) if the HA approach is suitable for GIA investigations and (2) if GIF results based on the HA approach are reliable in view of GIF activation due to GIA.

- Before we begin the comparison, it is beneficial to briefly repeat some background knowledge of GIA, which occurs due to the lithospheric loading by the ice sheet. The Earth deforms in response to this loading: beneath the load the lithosphere moves downward and rebounds once the load is gone. During subsidence the mantle flows away under the load and moves back once the ice sheet melts and the lithosphere is rebounding. Due to the viscous behaviour of the mantle, the process contains
- a time-independent elastic component and a time-dependent viscoelastic component, which delays the achievement of the state of equilibrium. The deformation of the lithosphere and mantle during the GIA process is related to the size of the ice sheet and this deformation has its peak value of sensitivity at a depth z (see Cathles, 1975, and Steffen et al., 2015 for a detailed derivation):

$$z \simeq \frac{1}{1.7\sqrt{\frac{1}{L^2} + \frac{1}{M^2}}},\tag{1}$$

- with L and M being the characteristic lengths of an elliptical ice sheet. A load size of 2000 km and 1500 km, for example, which is the north-south and east-west extension of the Fennoscandian Ice Sheet (Hughes et al., 2016), respectively, results in a peak value of sensitivity at 706 km depth. However, the depth with a half of the peak value gives a conservative estimate of how deep a load size can "see" into the mantle. The formula is similar to equation 1 except the factor 1.7 is replaced
  by 0.818, which gives a depth of 1467 km.
- by 0.818, which gives a depth of 1467 km.

The movement of lithosphere and mantle is also accompanied by stress changes. During loading (accumulation of ice) vertical and horizontal stresses are induced and during unloading (melting of ice) the vertical and horizontal stresses decrease. As soon as the unloading finished, the vertical stresses return to their value before the loading process started. However, as GIA is a viscoelastic

- process and stress migrates from the mantle into the lithospheric crust (Wu & Hasegawa, 1996), the horizontal stresses return much more slowly to the initial values. The change in stress with time is a major parameter for the determination of glacially induced earthquakes as stress calculations showed that the reactivation of pre-existing faults was induced by the melting of the ice sheet (Wu & Hasegawa, 1996; Johnston et al., 1998). Therefore, the stress distribution within GIF models
- has to be modelled correctly to allow an accurate analysis of former and current seismic hazards induced by glaciation and deglaciation.

Modelling these stress changes is however not straightforward using the FE method. Most FE software are based on engineering purposes and only the simple form of the equation of motion is solved (Wu, 2004):

$$90 \quad \nabla \cdot \mathbf{S}^{\mathbf{F}\mathbf{E}} = 0, \tag{2}$$

with  $S^{FE}$  as the stress tensor from FE software. To overcome this problem, Wu (2004) showed that the stress obtained from the FE modelling has to be transformed to GIA stresses:

$$\mathbf{S}^{\mathbf{GIA}} = \mathbf{S}^{\mathbf{FE}} + \rho_0 g_0 u_z \mathbf{I},\tag{3}$$

with S<sup>GIA</sup> as the GIA stress tensor, ρ<sub>0</sub> and g<sub>0</sub> as the density and gravity for the initial background
state, and u<sub>z</sub> as the vertical component of the displacement vector, to fulfil the simplified GIA equation for a flat Earth:

$$\nabla \cdot \mathbf{S^{GIA}} - \rho_0 g_0 \nabla u_z = 0 \tag{4}$$

(see Wu, 2004, and Steffen et al., 2014a, for a detailed derivation).

In the following, the predicted displacement and stress behaviour from the HA are compared with those from the WU approach for an earth model without a fault in order to compare the GIA contributions only. The next section introduces the two approaches and two-dimensional (2D) model setups. This is followed by a first test in sections 3 and 4, where a synthetic ice model and parameters following the study by Hetzel & Hampel (2005) are used. A second test in section 5 will show the displacement behaviour of a realistic ice load for both approaches in three dimensions (3D), using

the material parameters and horizontal dimensions of results used in Steffen & Kaufmann (2005) and Steffen et al. (2006), but keeping the boundary conditions and vertical dimensions of the specific methods.

## 2 Model description

We describe both approaches focusing on the GIA model only and do not include fault geometries.
Additionally, we will describe the synthetic ice model used in the comparison. The FE modelling is carried out using the software ABAQUS (Hibbitt et al., 2014).

#### 2.1 WU model

The WU model follows the approach developed by Wu (2004). The earth model has a thickness of 2891 km, from the surface of the Earth to the core-mantle boundary (Fig. 1(a)). Four layers are in-

- cluded in the model: the crust, the lithosphere of the mantle, the upper mantle, and the lower mantle. Each layer is described by density, Young's modulus and Poisson's ratio. Viscosity is applied to the lithospheric, upper and lower mantle only (for values see Fig. 1). At each boundary with density contrast, foundations are applied, which account for the restoring buoyancy force that drives GIA (see Wu, 2004). The model should have a width of at least 10 times of the ice-sheet size to avoid
- boundary effects, and the sides of the models are fixed in the horizontal direction. Quadrilateral plane strain elements are used (CPE4).

Figure 1

#### 2.2 HA model

The HA model follows the approach presented in Hetzel & Hampel (2005). We adopt the same model parameters as used in Hampel et al. (2009, see Fig. 1(b)). The earth model is 100 km thick,

from the earth surface to the lithosphere-asthenosphere boundary. The upper and lower mantle are not included in the model. Density, Young's modulus and Poisson's ratio are used for the crustal and lithospheric layer. A viscosity of  $1 \cdot 10^{23}$  Pa·s is used for the lithospheric mantle. At the bottom

- of the lithosphere, a lithostatic pressure and elastic foundations are applied. In addition, in e.g. Hetzel & Hampel (2005) dashpots were used as well, to represent the mantle viscosity, but were excluded in the 2D model presented in Hampel et al. (2009). Within the synthetic test, dashpots will not be used. However, we will show in section 5 that the inclusion of dashpots has negligible effects on the HA results. Additionally, the model is loaded with geostatic stresses to obtain a background
- stress state to simulate the advection of pre-stress. Due to the geostatic loading, the entire model deforms vertically. This deformation is constant in the entire model and can be subtracted from the displacement results of the ice loading process. The width of the model is 3000 km; hence, larger than the models in Hampel et al. (2009). The vertical sides of the model are fixed in horizontal directions. Triangular plane strain elements are used (CPE3; Hampel et al., 2010a).
- A study by Schotman et al. (2008) compared the displacement between the WU and HA models, i.e. "implementing the viscosity of the asthenosphere by dashpots instead of a finite element layer" (Hampel et al., 2009). A difference of less than 10% was obtained for the "modeled amount of flex-ure and rebound" (Hampel et al., 2009). However, Hampel et al. (2009) may have misunderstood the model setup and results of Schotman et al. (2008), who actually did not compare the WU and
- HA approach, but rather modified the WU approach by substituting the lower mantle with dashpots. This results in several differences. First the dashpots were not used at the bottom of the lithosphere in Schotman et al. (2008), but instead at the bottom of the 670 km boundary. This is in contrast to the dashpots used in HA models, which are always applied on the bottom of the lithosphere and this varies depending on the study between 80 and 120 km. Second, the upper mantle was included in
- the study by Schotman et al. (2008), which is not used in HA models. Third, Schotman et al. (2008) applied foundations at each layer with density contrast following Wu (2004) and thus avoided the implementation of lithostatic pressure and geostatic stresses. Schotman et al. (2008) conclude that they cannot use dashpot elements to replace the lower mantle as it leads to unacceptable errors for several computed parameters such as geoid height perturbation and horizontal velocities. Vertical
- deformation at the surface differed by up to 10%, while in 670 km depth the difference is up to 14%. Note again that this is for substituting the lower mantle only with dashpots and not for the whole mantle as generally done in the HA models as well as for foundations applied at all layers with density contrasts.

#### 2.3 Ice model

All models in sections 3 and 4 are loaded with a 200 km wide and 500 m thick ice sheet. Such ice load affects the Earth's subsurface to a depth of approximately 173 km (using equation 1 with a factor of 0.818 instead of 1.7). The amplitude of the ice load increases to its maximum value over

20 ka and decreases to zero in the following 10 ka. The time increment is 500 a. The load is modelled as pressure in the software ABAQUS (Hibbitt et al., 2014).

#### 165 3 Comparison of the displacement

The vertical displacement is obtained for both model approaches and only the geostatic displacement in the HA model is subtracted (0.64 m) from the FE modelling results. Fig. 2 shows the vertical displacement at three different locations at the surface of the model: at the centre of the model (0 km), at the ice margin (100 km) and 400 km away from the margin (500 km). Results at selected time steps

are also listed in Table 1.

Figure 2

Table 1

The HA model shows a gradual subsiding of the crust beneath the ice sheet during loading to -78.1 m at 0 km and -62.9 m at 100 km (Fig. 2). This is followed by an instantaneous elastic response as soon as deloading starts. At 30 ka (end of deloading) the vertical displacement is only -2.5 m at 0 km and -1.3 m at 100 km, and the uplift rate changes from 7.6 m/ka to 0.02 m/ka at the centre. At a location outside of the ice sheet, the vertical displacement increases up to 2.1 m and decreases with the start of deloading to the end of deloading to 0.4 m.

The vertical displacement obtained from model WU is smaller beneath the ice sheet (Fig. 2), but larger at the third location (500 km). During loading, the crust subsides to -53.0 m at 0 km and -40.9 m at 100 km. The subsidence is not linear during loading due to the viscoelasticity of the man-

- tle. Maximum displacement is also not directly at maximum glaciation, but within this example 1 ka later. The deloading process is accompanied by a slow uplift of the crust. The vertical displacement increases to -19.4 m and -15.7 m at 0 km and 100 km, respectively, at the end of deglaciation. After deloading, the uplift is still ongoing and the uplift velocity changes from 5.3 m/ka to 3.6 m/ka. At 500 km, the vertical displacement increases during loading to 4.7 m and decreases during and after
- deloading back to 0 m, indicating that this location is within the peripheral bulge. The subsiding of the crust at this location is not instantaneous, and a delay of 2 ka is observed.

The vertical displacement shows large differences (25.7 m at maximum glaciation) between the models HA and WU. Whereas the WU model shows a viscoelastic response of the earth model to the loading and unloading of the ice model, the HA model shows almost exclusively an elastic response.

We refer this difference to the different model dimensions in depth and thus the missing mantle layers. An ice load of 200 km width, has its peak value in sensitivity at a depth of 83 km, half of the peak value is reached at a depth of 173 km and a quarter still at 223 km. The model depth of only

100 km depth can therefore not be recommended when displacement changes due to the viscoelastic nature of the GIA process are calculated. This naturally alters the stress distribution and its change (stress migration), which we will investigate next.

## 4 Comparison of the stress

The horizontal stresses of the HA and WU models are shown in Fig. 3. Stresses from the WU model were transformed according to the equations described above. However, the stresses from HA models are not changed as the unmodified stresses ( $S^{FE}$ ) are used by Hetzel & Hampel (2005)

within the fault modelling steps. In contrast, Steffen et al. (2014a) use the modified stresses (S<sup>GIA</sup>) for their fault modelling. The vertical stresses of both approaches are almost identical (see Table 1) as this stress is induced by the load only.

#### 4.1 Horizontal stress

The horizontal stress is a function of the size of the ice model, the earth model parameters, and the 210 viscous behaviour of the mantle due to the stress migration from mantle to lithosphere. For a comparison of the horizontal stress results, the change with depth for two different time points is shown for both models (Fig. 3). The first time point is at maximum glaciation (20 ka, Fig. 3(a), (b)). The general distribution of the horizontal stress at glacial maximum is similar between WU and HA; however, the stress magnitude differs between both methods. While HA models reach an amplitude of almost

- 215 -10.3 MPa below the ice sheet within the crust, models by WU show only values of -8.4 MPa. At the bottom of the lithosphere (at 100 km), larger stress magnitudes are also obtained for HA models (13.8 MPa) compared to WU (10.8 MPa). The second time point used in the comparison shows larger differences within the stress magnitudes and the general distribution (Fig. 3(c), (d)). This time point corresponds with the end of deglaciation, and therefore reflects the time, when most known GIFs
- got reactivated. While models by HA show only a stress magnitude of -0.2 MPa below the ice sheet at a depth of 5 km (Table 1), WU models reach -2.1 MPa. A difference of 1.9 MPa is able to change the potential of a GIF from active to inactive or vice versa as well as the magnitude of the earthquake.

Figure 3

## 225 4.2 Differential stress

The differential stress is of high importance in stress field analyses. Models by HA show an instantaneous increase in 5 km depth in the stress magnitude during glaciation and an instantaneous decrease during deglaciation (Fig. 4). At the end of deglaciation, the rate of the stress magnitude changes significantly. The behaviour of the stress magnitude curve is similar to the curve of the ver-

230 tical displacement by HA, dominated by the elastic behaviour of the model. In contrast, the models

by WU show a viscoelastic behaviour, similar to the vertical displacement of WU (Fig. 4). At the end of deglaciation, the differential stresses are larger at all locations, which would favour a reactivation of a GIF at those time points.

Figure 4

#### 5 Comparison for a realistic ice sheet

The results obtained above indicate that the modelling approach from Hetzel & Hampel (2005) is not able to capture the displacement and horizontal stresses accurately using a synthetic ice sheet. A comparison of their modelling approach to observed datasets (e.g. relative sea level, GPS or gravity)

- has not been demonstrated so far. As such comparison is an important tool to properly validate the modelling approach, the results of a model based on the modelling approach by Wu (2004) and Hetzel & Hampel (2005) are compared to relative sea level (RSL) data as well as observed Global Navigation Satellite System (GNSS) velocities using a 3D flat-earth model with realistic ice history. The WU model has again a thickness of 2891 km using four layers (crust, lithospheric mantle,
- upper mantle, lower mantle) and a model width of 130,000 km is used to avoid boundary effects (Steffen et al., 2006). The HA model has a thickness of 120 km, which is a typical lithospheric thickness estimated from observations for Fennoscandia (Steffen & Kaufmann, 2005; Steffen & Wu, 2011) and used in the WU model as well. The width of the HA model is the same as for the WU model. Both models are meshed using brick elements (C3D8). We tested two different subsets of
- the model, one without dashpots at the base of the model (thus base of the lithosphere) and one with dashpots, to show the effect of dashpots used in HA models. The dashpot-property value is set equal to the upper-mantle viscosity used in the WU model and is unlike that used in HA models where the viscosities are too low to be representative of the whole mantle (Steffen et al., 2015). The Fennoscandian ice sheet model RSES by Lambeck et al. (1998) is used.
- Two RSL curves are chosen based on their location. The first set of RSL data is from Ångermanland, which is close to the centre of rebound. The second RSL curve is in the Netherlands, hence, outside of the Fennoscandian Ice Sheet in the peripheral bulge area. The results show large discrepancies between the two modelling approaches. For the HA model the discrepancies between the predictions and the observations reach 123 m and 22 m for Ångermanland (Sweden) and Leeuwarden
- (Netherlands), respectively (Fig. 5). In contrast, the results following Wu (2004) match the observations better and the discrepancies are much smaller: 11 m and 5.2 m for Ångermanland (Sweden) and Leeuwarden (Netherlands), respectively (Fig. 5).

Figure 5

In addition, we see that the HA models with and without dashpots show no difference. Hence, there is no effect of the dashpots and thus of the viscoelasticity of the underlying mantle. We attribute this behaviour to the large foundation applied at the base of the lithosphere, which Hampel et al. (2009) calculate taking into account the whole density of the asthenosphere instead of the density contrast

contrast.

The observed uplift velocity of Fennoscandia reaches its maximum of 10 mm/a in the Gulf of Bothnia (Fig. 6(a)), which is also predicted by the WU model (Fig. 6(b)). In addition, the modelled horizontal velocity field shows in general a similar pattern as the GNSS observations. A perfect match is not possible with flat-earth FE models due to the horizontal boundary conditions and partly

due to the lack of sphericity (Schotman et al., 2008). In contrast, the velocity fields obtained from the models following the approaches by Hampel et al. (2009) and Hetzel & Hampel (2005) are about zero in the entire region (Fig. 6(c), (d)), and thus cannot capture the typical uplift signal. This is due to the very high viscosity  $(1 \cdot 10^{23} \text{ Pa} \cdot \text{s})$  in their lithospheric mantle so that its Maxwell time is of the order of 100 ka, thus it behaves almost exclusively elastically during the glacial cycle. Even if

the dashpots are present, they cannot fully represent the viscoelastic relaxation of the mantle and the upward migration of stress, which is a result of the high foundation applied at the base of the model.

Figure 6

#### 285 6 Conclusions

The GIA process plays an important role in the reactivation of pre-existing faults (GIFs). Hence, the modelling of GIFs must be based on the correct description of the GIA process in the models. Two different GIF modelling approaches, one based on Steffen et al. (2014a) and Wu (2004), and the other based on Hetzel & Hampel (2005), were compared for their displacement and stress behaviour due to GIA during a loading process neglecting the effect of a fault. In our first test, a synthetic ice sheet was applied. Differences in the vertical displacement of up to 25.7 m (49%) and in the

- differential stress magnitudes of up to 1.9 MPa (90%) were obtained between the approaches. The general behaviour of the models presented in the displacement curve shows also large discrepancies, as the model following Hetzel & Hampel (2005) indicates almost exclusively an elastic response to
- the ice load, whereas the model following Wu (2004) reveals its viscoelastic response. It should be noted that some slight viscoelastic behaviour in the deformation and stress changes from HA models is solely due to the viscoelastic lithosphere and not the mantle. Previous studies with HA models (Turpeinen et al., 2009; Hampel et al., 2009) showed viscoelastic displacements using low viscosity values of  $4 \cdot 10^{18}$  Pa·s for the lithosphere and  $1 \cdot 10^{19}$  Pa·s for the lower crust, which give Maxwell