# Peer review of "Comparison of the glacial isostatic adjustment behaviour in glacially induced fault models"

_Geoscientific Model Development, 2016_

## Referee Comment (RC1) · Anonymous Referee #1 · 16 Apr 2016

Review of manuscript "Comparison of the glacial isostatic adjustment behaviour in glacially induced fault models" by Steffen et al, submitted to Geosci. Model Dev. Discuss., 2016.

Dear Editor,

this work compares two GIA (glacial isostatic adjustment) models. The first (named WU) has been developed by the authors (Steffen et al., 2014), the second (HA) has been proposed by Heizel and Hampel in 2005. Both models have the purpose of describing the behaviour of faults in regions subject to deglaciation. Using various quantitative arguments, Steffen et al. aim at showing that the HA model is "not recommended" due to "poor performance" and argue that WU has "rigorous support" while HA has not. The authors conclude that "thorough modelling" of the GIA process is

a prerequisite to understand the faults response to changes in surface loads. Note that here, the movement of glacially induced faults is not explicitly modelled; rather the concern is about the GIA models setup.

In general, since the WU and HA models are indeed showing marked differences, I am not very surprised that their outcomes in terms of displacements and stresses differ. I also agree with the authors that thorough modelling is fundamental in order to obtain meaningful results, but this is certainly not new. I definitively cannot capture the usefulness of this study, which appears to have the narrow purpose of showing the limitations of the HA approach (which certainly exist, as in any modelling approaches). In the recent past, there has been a robust discussion between the two groups (see DOI 10.1002/2013TC003450 and DOI 10.1002/2014TC003772). During that discussion, some of the points presented in the present work have already been made, like the one about the suitability of the HA methodology for GIA studies. Furthermore, the discussion has also gone into details about the differences between the WU and HA models, which is also one of the purposes of present study.

My general impression (but of course I might be wrong) is that the present work is mostly aimed to reignite the debate between the two groups, without clearly identifiable benefits for the Geosci. Model Dev. community.

---

## Author Comment (AC1) · 30 Apr 2016

Dear Reviewer,

We thank you for reviewing our paper and your feedback.

First, we would like to address your last concern that our paper is "without clearly identifiable benefits for the Geosci. Model Dev. community". The Aims and Scope of Geoscientific Model Development, see website, list several types of papers to be considered for publication, one type being "Model evaluation papers", which aim for "full evaluations of previously published models". Our paper clearly matches this type of definition as, e.g., it says "Typically, this comprises a comparison of the performance of different model configurations or parameterisations." We present a comparison of already existing models and test them with respect to their model configuration and the

different outputs with respect to stresses and displacement. Therefore, we kindly rebut your concern.

Our second point discusses the usefulness of our study, which is questioned by you, especially as you correctly noted that a comment and a reply were recently published discussing the two modelling approaches. We would like to note here that our reply only dealt with rebutting the many issues raised in the comment, but not a detailed comparison of the two approaches, which is, on another note, not possible due to page limitations for a reply by the other journal. Therefore, the reply paper contains only one figure which adds information to a figure by Hampel et al. (2009) that was mentioned in the comment to support one of the issues raised. The reason to submit the current manuscript coincides with your statement that you were "not very surprised that their outcomes in terms of displacements and stresses differ". It is of course clear that two different approaches can lead to different results. However, the questions are how much they differ and if the process they want to describe is actually addressed by the approach. A side-by-side comparison of model results including a fault is not possible due to several reasons and this was already discussed in the reply. However, as both approaches are claimed by each author-team to show the response of faults during a glacial cycle, the correct description of the glacial isostatic adjustment (GIA) behavior is a necessity, and this has only been touched in the reply with a figure of a displacement curve. Here we show by adding substantial material compared to that of the reply (2D and 3D models are tested, the stress behavior is shown, different locations and GIA observations are addressed) that one approach unfortunately fails to do so. This may not come as a surprise for someone with GIA modelling background, but researchers not working in such field may be surprised. It is not our aim to further re-ignite the discussion between the two groups, it is more important for us to show researchers interested in these studies (GIA, tectonics, intraplate seismicity and more) that the base for the analysis of fault response due to large-scale load scenarios of ice and water must also give correct description of other GIA observables.

We are very thankful for your review as it shows that our motivation for the paper has to be refined. Hence, we will adjust our paper, mainly in the introduction, according to your concerns to make our points clearer and more concise as well as to show the importance of our study especially with respect to the former published discussion.

—————————————————————

---

## Referee Comment (RC2) · Anonymous Referee #2 · 13 May 2016

Authors of the manuscript "Comparison of the glacial isostatic adjustment behaviour in glacially induced fault models" present a comparative study of two approaches to modeling of the glacial isostatic adjustment (GIA). As I've understood from the text, it is a continuation of debates between two groups of authors according reliability of methods they use.

In the introductory part (55, p.2) authors declare that the aim of his study is to compare two approaches ( abbreviated as "WU" and "HA") based on benchmarking of two typical setups from the previous studies. However, authors also notes that they aim "... to verify (1) if the HA approach is suitable for GIA investigations and (2) if GIF results based on the HA approach are reliable in view of GIF activation due to GIA". This phrase might indirectly point on prejudice of the authors that won't pass for a comparative benchmarking study.

[Figure]

Main complaints of the authors concerning the HA models are: (i) the HA model does not take into account viscosity of the whole mantle using the dashpot approach instead, (ii) the HA models neglect an effect of the free surface and topography of the boundaries between internal layers caused by loading (as far as I understood from Eq.3 and Eq. 4) , and (iii) the bottom boundary condition (10, p.1). I'm totally agree with the authors that the using more realistic lithospheric structure and boundary conditions allow for more natural behaviour of the model. It is obvious and do not needs additional proofs. However, it makes numerical models unreasonably complex in some cases. That is why we use often such a simplified approaches like the WU and HA and both of them might be valid under certain conditions or not, and that is why I can not support the conclusion of the manuscript that the HA modeling approach is unappropriate.

In my opinion, comparison of two different approaches with the same model parameters like the mantle viscosity is unacceptable because of different limitations for both models. For instance, the WU model fits to observed data (Fig. 5) only for the particular viscosity structure but it does not mean that the used values of viscosity correspond to real ones. There are plenty of published radial viscosity profiles based on GIA studies and the geoid inversion. Using any of that within, say, WU or HA approach gives sufficiently different response on surface loading as well as including in the model such important factors as dependance of viscosity both on stress and temperature, changing of elastic thickness of the lithosphere under loading, compressibility, dynamic pressure caused by convection in the mantle due to inhomogeneous density structure, etc.

A conclusion that " we see that the HA models with and without dashpots show no difference" is also very strange to me. Let's consider an end-member example: a dashpot with infinite viscosity (fixed boundary). It must change the solution, otherwise calculations are wrong.

I realize, that the objective and impersonal benchmarking study that includes at least three different approaches is really important for further advances in the GIA modeling, but not a criticism of the particular modeling approach (HA) that I can see from the

manuscript.

---

## Author Comment (AC2) · 12 Jun 2016

Dear Reviewer,

We thank you for reviewing our paper and your feedback. Please find below our answers to your comments.

"In the introductory part (55, p.2) authors declare that the aim of his study is to compare two approaches (abbreviated as "WU" and "HA") based on benchmarking of two typical setups from the previous studies. However, authors also notes that they aim "... to verify (1) if the HA approach is suitable for GIA investigations and (2) if GIF results based on the HA approach are reliable in view of GIF activation due to GIA". This phrase might indirectly point on prejudice of the authors that won't pass for a comparative benchmarking study."

We will revise the introduction to remove any prejudging content.

"I'm totally agree with the authors that the using more realistic lithospheric structure and boundary conditions allow for more natural behaviour of the model. It is obvious and do not needs additional proofs. However, it makes numerical models unreasonably complex in some cases. That is why we use often such a simplified approaches like the WU and HA and both of them might be valid under certain conditions or not, and that is why I can not support the conclusion of the manuscript that the HA modeling approach is unappropriate."

We disagree, as we did not say the HA approach is inappropriate. We state that the HA approach cannot be recommended for typical GIA loads that are continental-scale ice sheets which require the inclusion of the whole mantle in the GIA model. In turn, we further state that the approach may work for load dimensions of <100 km diameter, thus, in a line with your comment that the HA approach may indeed be valid under this special condition. In addition, the aim of our study is not the inclusion of realistic lithospheric structure, but the presentation of the effect of neglecting the mantle in GIA models.

"In my opinion, comparison of two different approaches with the same model parameters like the mantle viscosity is unacceptable because of different limitations for both models. For instance, the WU model fits to observed data (Fig. 5) only for the particular viscosity structure but it does not mean that the used values of viscosity correspond to real ones. There are plenty of published radial viscosity profiles based on GIA studies and the geoid inversion. Using any of that within, say, WU or HA approach gives sufficiently different response on surface loading as well as including in the model such important factors as dependance of viscosity both on stress and temperature, changing of elastic thickness of the lithosphere under loading, compressibility, dynamic pressure caused by convection in the mantle due to inhomogeneous density structure, etc."

Thank you for your opinion. First, we would like to note that it is not only the mantle

viscosity (or better say the mantle viscosity profile) which needs to be tuned for a good fit of GIA model results to observations. There is trade-off between lithospheric thickness and the viscosity in the asthenosphere below, so that altering one may change the other. We are certainly aware of the different viscosity profiles in the literature, but we cannot see a benefit of presenting results of other profiles which may give a worse fit to the observations for both the WU and HA approach. You made a good point that other important factors which may have an effect have not been taken into account in both approaches, however, this is not the goal of our manuscript – and therefore we focus on the two approaches as published to date. We will include this information in the revised manuscript. However, we disagree with your comment that both model approaches cannot be compared to each other. The HA model replaces the mantle with a boundary condition and as the approach is intended to model GIF movement, it is also modelling the GIA behavior. Therefore, both approaches are indeed comparable with respect to the GIA response, which is done in this study. In addition, models need to be compared to each other as they are used in their respective studies. A modification of one approach is not the goal and purpose of a benchmarking study. We can only repeat once more that both approaches (HA and WU) were according to their authors designed to explain the physics behind the motion of glacially induced faults, and a prerequisite for that is a correct description of the GIA behavior. We can assign different viscosity profiles to the HA or WU model, but we will always arrive at the conclusion that the HA approach cannot sufficiently describe the well-known GIA behavior for large-scale load scenarios. We will make our point clearer in the revised manuscript.

"A conclusion that "we see that the HA models with and without dashpots show no difference" is also very strange to me. Let's consider an end-member example: a dashpot with infinite viscosity (fixed boundary). It must change the solution, otherwise calculations are wrong."

In case the boundary conditions are correctly applied, the last sentence is a true statement. We have therefore tested the HA approach with the two different boundary conditions used by HA according to publications, i.e. with and without dashpots. As there are no differences, we arrived at the conclusion that the boundary conditions are imperfect. Therefore, an end-member example as suggested will not change the solution. We refer this strange behavior to the large foundation applied at the bottom. On another note, Schotman et al. (2008) calculated the displacement using infinite elements instead of dashpots at the 670 km boundary, and, as expected, differences were obtained. However, the foundation prescribed at this boundary was small and is not comparable to the values as used within HA models. Hence, changes at the bottom of the model definitely change the solution as long as the boundary condition allows it at all. To verify our answer, we have tested the HA and WU model approach using upper mantle viscosities of 4*1018 Pas and 4*1022 Pas. The RSL as well as GPS uplift rates are shown in the Figures 1 to 4. Please note that the viscosity used in the benchmark study is 4*1020 Pas. The figures show that a change in viscosity has no effect on the behavior of the HA model due to the large foundation value. In contrast, the WU model shows large changes in the displacement field and no fit to the observations can be obtained. The low viscosity of 4*1018 Pas leads to a very short Maxwell time and the lithosphere rebounds back quickly and almost no rebound is left today. The higher viscosity of 4*1022 Pas refers to a larger Maxwell time and the lithosphere cannot sink into the mantle. Therefore, the land uplift (Fig. 2) is small. We have added Fig. 1 and 2 to the manuscript.

"I realize, that the objective and impersonal benchmarking study that includes at least three different approaches is really important for further advances in the GIA modeling, but not a criticism of the particular modeling approach (HA) that I can see from the manuscript."

We are sorry that we do not understand this comment. Our manuscript deals with the comparison of two published approaches for GIF modeling. To our best knowledge there is no third (or fourth. . .) approach for GIF modeling published so far, which can

and should be benchmarked regarding the GIA behavior with the WU and the HA approach.

Please also note the supplement to this comment:
http://www.geosci-model-dev-discuss.net/gmd-2016-43/gmd-2016-43-AC2-supplement.pdf

───────────────────────────────

[Figure]

**Fig. 1.** Land uplift curves using an upper mantle viscosity of 4*1018 Pas. For more information please see Figure 5 in the benchmarking paper.

[Figure]

**Fig. 2.** Land uplift curves using an upper mantle viscosity of 4*1022 Pas. For more information please see Figure 5 in the benchmarking paper.

[Figure]

**(a) Observation (Kierulf et al. 2014)**

**(b) 4E20 Pa s**

**(c) 4E18 Pa s**

**(d) 4E22 Pa s**

**Fig. 3.** Uplift rates of the HA model approach for different viscosities.

[Figure]

[Figure]

**Fig. 4.** Uplift rates of the WU model approach for different viscosities. Please note the different color bars for the top and bottom rows.

**Supplement:**

Manuscript prepared for Geosci. Model Dev.
with version 2015/04/24 7.83 Copernicus papers of the LATEX class copernicus.cls.
Date: 12 June 2016

**Comparison of the glacial isostatic adjustment behaviour in glacially induced fault models**

Rebekka Steffen[1], Holger Steffen[2], and Patrick Wu[3]

[1]Department of Earth Sciences, Uppsala University, Villavägen 16, 75692 Uppsala, Sweden
[2]Lantmäteriet, Lantmäterigatan 2C, 80182 Gävle, Sweden
[3]Department of Earth Sciences, The University of Hong Kong, Pokfulam Road, Hong Kong

*Correspondence to:* Rebekka Steffen (rebekka.steffen.geo@gmail.com)

**Abstract.** We compare the glacial isostatic adjustment (GIA) behaviour of two approaches developed to model the movement of a glacially induced fault (GIF) as a consequence of stress changes in the Earth's crust caused by the GIA process. GIFs were most likely, but not exclusively reactivated at the end of the last glaciation. Their modelling is complicated as the GIA process involves different spatial and temporal scales and they have to be combined to describe the fault reactivation process accurately. Model approaches have been introduced by Hetzel & Hampel (2005, termed HA in this paper) and Steffen et al. (2014a, termed WU in this paper). These two approaches differ in their geometry, their boundary conditions and the implementation of stress changes. While the WU model is based on GIA models and thus includes the whole mantle down to the core-mantle boundary at a depth of 2891 km, the HA  model includes only the lithosphere (mostly 100 km) and  simulates the mantle using dashpots.  It further applies elastic foundations and a lithostatic pressure at the base of the lithosphere, while the WU  model applies elastic foundations at all horizontal boundaries in the model with density contrasts. Using a synthetic ice model as well as the Fennoscandian Ice Sheet, we find large discrepancies in modelled displacement, velocity and stress between these approaches.  Assuming a typical viscosity profile from GIA studies, the HA model has difficulties in explaining relative  sea-level curves in Fennoscandia such as the one of Ångermanland (Sweden),  while the WU model only differs within the error bar of  the observations. In addition, the HA model cannot predict the typical velocity field pattern in Fennoscandia. As we also find prominent differences in stress, we conclude that the simulation of the mantle using dashpots is not recommended for modelling the GIA process of continental-scale ice sheets. The earth model should consist of both lithosphere and mantle, in order to correctly model the displacement and stress changes during GIA. We emphasize that a thorough modelling of the GIA process is a prerequisite before conclusions on understanding GIF evolution can be drawn.

**1 Introduction**

Geodynamic models are developed to advance our understanding of the many individual as well as overlapping processes of the Earth. A common phenomenon is that several models co-exist for the same process and they should be compared or benchmarked in order to verify that each method works correctly. Benchmark studies thus have been performed for dedicated convection models (e.g. Zhong et al., 2008; Tosi et al., 2015), dynamo models (e.g. Christensen et al., 2001; Jackson et al., 2014) or models of glacial isostatic adjustment (GIA; Spada et al., 2011). The latter describe the response of the Earth in terms of deformation as well as stress, rotation and geopotential changes due to changing ice-ocean load distributions on the Earth's surface. Among other things, the GIA model benchmark showed that the displacement results from models based on the viscoelastic normal mode method are comparable to results from spectral-finite element and finite-element (FE) models, when an earth model is subjected to an ice load. This is of importance as FE models are able to handle faults and lateral heterogeneities in the Earth's subsurface as well as nonlinear or composite rheologies in the mantle.

In this study, our focus is on the GIA description in glacially induced fault (GIF) models. GIFs represent reactivated faults in or nearby formerly glaciated areas such as North America or northern Europe (e. g. Kujansuu, 1964; Lagerbäck, 1978; Quinlan, 1984; Johnston, 1987; Olesen, 1988; Dyke et al., 1991; Shilts et al., 1992; Fenton, 1994; Arvidsson, 1996; Muir-Wood, 2000; Stewart et al., 2000; Munier & Fenton, 2004; Sauber & Molnia, 2004; Lagerbäck & Sundh, 2008; Brandes et al., 2012). Even historical earthquakes of the last 1200 years in northern Germany are related to the last glaciation of northern Europe (Brandes et al., 2015). Movement of faults under the ice sheets in Laurentia and Fennoscandia was suppressed during glacial loading (Johnston, 1987), but was reactivated near the end of deglaciation (Wu & Hasegawa, 1996).

GIF modelling has been a challenging task as it involves the large spatial scale (> 1000 km) and long time scale  tectonic stress (millions of years), the  GIA-induced stress (thousands of years) and the short-term earthquake motion (a few seconds to minutes) at a fault (of some km length). Nonetheless, two approaches for GIF modelling were introduced in recent years, and both used FE techniques: the first was presented by Hetzel & Hampel (2005, hereafter denoted as HA) based on rather geological aspects and the second by Steffen et al. (2014a) based on the GIA modelling approach by Wu (2004, hereafter WU), which was part of the benchmark study of Spada et al. (2011). Hence, WU has rigorous support from other GIA modelling techniques, while HA has not so far, although it was applied in numerous GIF, but mainly parameter studies (Hampel & Hetzel, 2006; Hampel et al., 2007, 2009; Turpeinen et al., 2009; Hampel et al., 2010a, b). Recently, Steffen et al. (2015) indicated that there are large discrepancies in the vertical deformation behaviour between the two approaches, whose reasons have not been fully investigated yet. Moreover, a comparison of horizontal deformation and stress has not been done as well as a test of the HA approach with a more realistic, continental-scale ice model which has often been

used in GIA investigations. Therefore, our aim in this study is to compare these two approaches in terms of their description of the GIA process to verify (1)  65  the GIA response of both approaches by comparison to a set of typical GIA observations, i. e. relative sea-level data and the velocity field, and (2) if GIF results based on  both approaches are reliable in view of GIF activation due to GIA. A correct description of the GIA behaviour is a necessary foundation of GIF models, and is especially important with respect to seismic hazard assessments due to GIFs as their response is due to GIA.

70      Before we begin the comparison, it is beneficial to briefly repeat some background knowledge of GIA, which occurs due to the lithospheric loading by the ice sheet. The Earth deforms in response to this loading: beneath the load the lithosphere moves downward and rebounds once the load is gone. During subsidence  mantle rocks flow away under the load and  move back once the ice sheet melts and the lithosphere is rebounding. Due to the  viscoelastic behaviour 75 of the mantle, the process contains a time-independent elastic component and a time-dependent  viscous component, which delays the achievement of the state of equilibrium. The depth of the deformation of the  GIA process is related to the size of the ice sheet and  peaks at a depth $z$ (see Cathles, 1975, and Steffen et al., 2015 for a detailed derivation):

$$\quad z \simeq \frac{1}{1.7 \sqrt{\frac{1}{L^2} + \frac{1}{M^2}}}, \qquad (1)$$

with $L$ and $M$ being the characteristic lengths of an elliptical ice sheet. A load size of 2000 km and 1500 km, for example, which is the north-south and east-west extension of the Fennoscandian Ice Sheet (Hughes et al., 2016), respectively, results in a peak value of sensitivity at 706 km depth. However, the depth with a half of the peak value gives a conservative estimate of how deep a load size can 85 "see" into the mantle. The formula is similar to equation 1 except the factor 1.7 is replaced by 0.818, which gives a depth of 1467 km. Thus, mantle material at a depth of at least 1500 km is affected due to the Fennoscandian Ice Sheet. Any FE model that tries to model GIA-induced displacement and stress for comparison with GIA observations (e. g. relative sea-level, GPS or gravity data) needs to account for the movement of mantle material at those depths either by using the dedicated boundary 90 conditions or by extending the model to at least this depth.

[revised manuscript text omitted]

To verify this observation, other viscosities were applied at the dashpots in the HA model and

in the upper mantle of the WU model. An upper mantle viscosity of $4 \cdot 10^{18}$ Pa·s is used in a first

test. This low value implies a shorter Maxwell relaxation time, which leads to a rapid rebound of the lithosphere. The lithosphere shows an almost instantaneous reaction to changes in the ice load. This is visible for the WU and HA models in Fig. 7; however, nothing has changed for the HA model when compared to results using a viscosity of $4 \cdot 10^{20}$ Pa·s for the dashpots. In a second test, a higher

310 viscosity of $4 \cdot 10^{22}$ Pa·s is used for the dashpots in the HA model as well as upper mantle viscosity in the WU model. If the material has a high viscosity, the Maxwell relaxation time is also increased. This implies that the mantle is too stiff and prevents sinking of the lithosphere into the mantle. Only for longer time than the Maxwell relaxation time the mantle will behave viscoelastically and deform. This results in a small deformation during loading and hence, a small amount of land uplift during

315 and after deglaciation. While the WU model shows exactly this behaviour in Fig. 8, the HA model shows no variation in Figs. 5 and 7 due to viscosity variations for the dashpot. This implies that any viscosity could be used for the dashpot, as the foundation value is set too high, and the dashpots cannot be used to model the GIA process.

320    Figure 7

       Figure 8

**6   Conclusions**

325 The GIA process plays an important role in the reactivation of pre-existing faults (GIFs). Hence, the modelling of GIFs must be based on the correct description of the GIA process in the models. Two different GIF modelling approaches, one based on Steffen et al. (2014a) and Wu (2004), and the other based on Hetzel & Hampel (2005), were compared for their displacement and stress behaviour due to GIA during a loading process neglecting the effect of a fault. In our first test, a synthetic ice

330 sheet was applied. Differences in the vertical displacement of up to 25.7 m (49 %) and in the differential stress magnitudes of up to 1.9 MPa (90 %) were obtained between the approaches. The general behaviour of the models presented in the displacement curve shows also large discrepancies, as the model following Hetzel & Hampel (2005) indicates almost exclusively an elastic response to the ice load, whereas the model following Wu (2004) reveals its viscoelastic response generally known from

335 former GIA studies. It should be noted that some slight viscoelastic behaviour in the deformation and stress changes from HA models is solely due to the viscoelastic lithosphere and not the mantle. Previous studies with HA models (Turpeinen et al., 2009; Hampel et al., 2009) showed viscoelastic displacements using low viscosity values of $4 \cdot 10^{18}$ Pa·s for the lithosphere and $1 \cdot 10^{19}$ Pa·s for the lower crust, which give Maxwell relaxation times of the order of ten years. Thus they relax too fast

340 and too early to be of importance to the triggering of GIF movement by the GIA process.

Applying a realistic ice sheet and using the same vertical dimensions of each modelling approach presents a good fit to RSL and GPS observations for the model after Wu (2004), but leaves large differences in the model of Hetzel & Hampel (2005). The uplift velocities predicted by their approach exhibit no significant uplift today in entire northern Europe due to the last glaciation. As the method after Wu (2004) was recently benchmarked to GIA models using the commonly used viscoelastic normal-mode method (Schotman et al., 2008; Spada et al., 2011) and performing excellently there, we suggest that this approach is preferable when simulating GIFs, i. e. if continental-scale ice sheets are taken into account. Parameter tests for GIF with this method can be found in Steffen et al. (2014b, c). However, both approaches are also limited with respect to other parameters, as temperature dependence, change in elastic thickness of the lithosphere during loading, convection in the mantle as well as realistic density structures are not included, which might effect the displacement and stress changes induced by GIA. Nevertheless, the importance of the mantle is shown here, which is a major parameter in GIA models.

Unfortunately, the approach by Hetzel & Hampel (2005) cannot be recommended due to their poor performance for GIF studies due to the results in GIA investigations obtained within this study, which are a major prerequisite for GIF analysis. Moreover, our comparison to GIA observations questions all results of earlier studies applying this approach. The approach by Hetzel & Hampel (2005) may only be feasible if the load (e. g. ice or water) is small enough to reach maximum sensitivity in the lithosphere. Steffen et al. (2015) showed that this is possible for load dimensions of <100 km diameter; however, all studies to date applying the HA approach used partly much larger loads, and hence these results have to be treated with care..

**Code availability**

The input files of the HA and WU model using a synthetic ice sheet are included in the supplementary material (HA.inp and WU.inp). The set-up of the HA model is obtained from Hampel et al. (2009, 2010a). The input files of the 3D models using a realistic ice sheet are available upon request, however, the structure of the input files is the same as for the HA.inp and WU.inp.

*Acknowledgements.* We are very grateful for the excellent reviews and numerous valuable suggestions from two anonymous referees and the Editor, Philippe Huybrechts, which have greatly improved this manuscript. We would like to thank Björn Lund and Peter Schmidt (Uppsala University) for several discussions about the modelling approaches as well as Hugo Schotman for details on the models used in Schotman et al. (2008).

Captions to Figures:

**Figure 1**:

480 Schematic sketch of the model structures following the approach by Wu (2004, (a)) and by Hetzel & Hampel (2005, (b)) used for the comparison of displacement and stress using a synthetic ice sheet. The width of the model is only changed when using the realistic ice sheet with 130,000 km for both model types. In addition, the lithospheric thickness is increased from 100 km to 120 km, when using a realistic ice sheet model. (b) is adapted from Hetzel & Hampel (2005).

485

**Figure 2**:

Vertical displacement at the surface for the model HA (red) and the model WU (blue). Three different locations are shown: beneath the centre of the ice sheet (0 km, solid line), at the boundary of the ice sheet (100 km, dashed line), and 400 km away from the ice sheet border (500 km, dotted line).

490 The increase and decrease of the load is shown in the upper part of the figure as purple curve.

**Figure 3**:

Horizontal stresses for two time points for model HA (left) and model WU (right). Upper row ((a) and (b)) is for glacial maximum (20 ka) and lower row ((c) and (d)) is for end of deglaciation (30 ka).

495 The size of the ice sheet is shown as purple bar on top of each model.

**Figure 4**:

Differential stress at 5 km depth of the model HA (red) and model WU (blue). Locations as in Fig. 2.

500 **Figure 5**:

Observed and modelled relative sea-level (RSL) curves for (a) Leeuwarden (Netherlands) and (b) Ångermanland (Sweden). Solid lines are modelled predictions using the RSES Fennoscandian Ice Sheet model (Lambeck et al., 1998) for models following the approach by Wu (2004, blue) with a depth of 2891 km, by Hampel et al. (2009, red) with a depth of 120 km and without dashpots at the

505 base of the model, and by Hetzel & Hampel (2005, green) with a depth of 120 km and using dashpots at the base of the model to simulate the viscosity of the upper mantle. An upper mantle viscosity of $4 \cdot 10^{20}$ Pa·s is sued. The observations from RSL data are shown in black, including the error in time and height.

510 **Figure 6**:

Observed and modelled velocities in northern Europe. (a) Global Navigation Satellite System (GNSS) observations from Kierulf et al. (2014). (b) - (d) Modelled velocities using the RSES Fennoscandian Ice Sheet model (Lambeck et al., 1998) for models following the approach (b) by Wu (2004) with a

depth of 2891 km, (c) by Hampel et al. (2009) with a depth of 120 km and without dashpots at the base of the model, and (d) by Hetzel & Hampel (2005) with a depth of 120 km and using dashpots at the base of the model to simulate the viscosity of the upper mantle.

**Figure 7**:
Same as Fig. 5, but an upper mantle viscosity of $4 \cdot 10^{18}$ Pa·s is used.

520

**Figure 8**:
Same as Fig. 5, but an upper mantle viscosity of $4 \cdot 10^{22}$ Pa·s is used.

(a)

[Figure]

(b)

**Figure 1.**

[Figure]

**Figure 2.**

[Figure]

**Figure 3.**

[Figure]

**Figure 4.**

[Figure]

**Figure 5.**

[Figure]

**Figure 6.**

[Figure]

**Figure 7.**

[Figure]

**Figure 8.**

**Table 1.** Selected results of the models with synthetic ice load. Vertical displacement at the surface and horizontal, vertical and differential stresses at 5 km depth at three locations and four different time points (10 ka - 50 % of glaciation, 20 ka - maximum glaciation, 30 ka - end of deglaciation, 40 ka - 10 ka after the end of deglaciation).

| | Model HA | | | Model WU | | |
|---|---|---|---|---|---|---|
| | 0 km | 100 km | 500 km | 0 km | 100 km | 500 km |
| Vertical displacement at 10 ka [m] | -38.6 | -31.2 | 1.0 | -21.8 | -16.6 | 1.6 |
| Vertical displacement at 20 ka [m] | -78.1 | -62.9 | 2.1 | -52.4 | -40.4 | 4.2 |
| Vertical displacement at 30 ka [m] | -2.5 | -1.3 | 0.4 | -19.4 | -15.7 | 1.7 |
| Vertical displacement at 40 ka [m] | -2.4 | -1.2 | 0.3 | -4.4 | -3.3 | -0.7 |
| Horizontal stress at 10 ka [MPa] | -4.0 | -2.0 | 0.5 | -3.2 | -1.5 | 0.4 |
| Horizontal stress at 20 ka [MPa] | -8.0 | -4.0 | 1.1 | -7.5 | -3.7 | 0.9 |
| Horizontal stress at 30 ka [MPa] | -0.2 | 0.0 | -0.1 | -2.1 | -1.3 | 0.3 |
| Horizontal stress at 40 ka [MPa] | -0.2 | 0.0 | -0.1 | -0.4 | -0.2 | -0.1 |
| Vertical stress at 10 ka [MPa] | -2.2 | -1.1 | 0.0 | -2.2 | -1.1 | 0.0 |
| Vertical stress at 20 ka [MPa] | -4.4 | -2.1 | 0.0 | -4.4 | -2.2 | 0.0 |
| Vertical stress at 30 ka [MPa] | 0.0 | 0.0 | 0.0 | 0.0 | 0.0 | 0.0 |
| Vertical stress at 40 ka [MPa] | 0.0 | 0.0 | 0.0 | 0.0 | 0.0 | 0.0 |
| Differential stress at 10 ka [MPa] | 1.8 | 1.5 | 0.5 | 1.0 | 1.1 | 0.4 |
| Differential stress at 20 ka [MPa] | 3.7 | 2.9 | 1.1 | 3.1 | 2.6 | 0.9 |
| Differential stress at 30 ka [MPa] | 0.2 | 0.1 | 0.1 | 2.1 | 1.3 | 0.3 |
| Differential stress at 40 ka [MPa] | 0.2 | 0.1 | 0.1 | 0.4 | 0.2 | 0.1 |